# Identifying and Predicting Healthcare Waste Management Costs for an Optimal Sustainable Management System: Evidence from the Greek Public Sector

**DOI:** 10.3390/ijerph19169821

**Published:** 2022-08-09

**Authors:** Anastasios Sepetis, Paraskevi N. Zaza, Fotios Rizos, Pantelis G. Bagos

**Affiliations:** 1Postgraduate Health and Social Care Management Program, University of West Attica, 12244 Athens, Greece; 2Department of Computer Science and Biomedical Informatics, University of Thessaly, 35131 Lamia, Greece; 3Department of Business Administration, University of West Attica, 12241 Athens, Greece

**Keywords:** climate change, public health, waste management, healthcare waste, medical waste, sustainability in healthcare, Greece

## Abstract

The healthcare sector is an ever-growing industry which produces a vast amount of waste each year, and it is crucial for healthcare systems to have an effective and sustainable medical waste management system in order to protect public health. Greek public hospitals in 2018 produced 9500 tons of hazardous healthcare wastes, and it is expected to reach 18,200 tons in 2025 and exceed 18,800 tons in 2030. In this paper, we investigated the factors that affect healthcare wastes. We obtained data from all Greek public hospitals and conducted a regression analysis, with the management cost of waste and the kilos of waste as the dependent variables, and a number of variables reflecting the characteristics of each hospital and its output as the independent variables. We applied and compared several models. Our study shows that healthcare wastes are affected by several individual-hospital characteristics, such as the number of beds, the type of the hospital, the services the hospital provides, the number of annual inpatients, the days of stay, the total number of surgeries, the existence of special units, and the total number of employees. Finally, our study presents two prediction models concerning the management costs and quantities of infectious waste for Greece’s public hospitals and proposes specific actions to reduce healthcare wastes and the respective costs, as well as to implement and adopt certain tools, in terms of sustainability.

## 1. Introduction

It is common truth that the quality of our environment affects the public health, and thus the healthcare sector should perform respective actions in order to preserve it [1]. The healthcare sector remains one of the largest industries globally due to the fact that health expenditures are calculated at around 10 percent of the global economic output [2], and it has an important role in damaging and degrading the natural environment due to the 24/7 operation, their constantly energy consumption [3], and the huge amounts of healthcare/medical wastes they produce [4]. Even though the healthcare sector is considered to be greener than other industries worldwide [5], its global carbon footprint is found to be at about 4.4 percent of the world’s total greenhouse emissions [6,7,8] and is expected to be tripled by 2050, reaching six gigatons a year [9].

Climate change is deemed as one of the most crucial threats in the last few decades, and many researchers, academics, experts, and other interested parties have expressed their concerns and given considerable attention to this matter [10,11]. The World Health Organization (WHO) has shown extreme concern regarding the high consumption of resources and the impact of climate change and the environment of the healthcare providers [12]. It is noteworthy that a study by the WHO in 2021 stated that 25% of total deaths worldwide were due to unhealthy work and living environment [13]. The Lancet stated that one of the biggest global health threats of the 21st century is the climate change, and to this end, the lives and well-being of billions of people are at an increased risk. [14]. Moreover, in a recent study, Lancet [15] stated that pollution (i.e., air pollution, water pollution, toxic occupational hazards etc.) was responsible for one in six deaths (almost 9 million premature deaths worldwide). In 2015, United Nations published the “Transforming our world: the 2030 Agenda for Sustainable Development”, which sets 17 Sustainable Development Goals regarding (among others) the preservation and protection of the natural resources and our planet, the environmental degradation, and the inequalities within and among countries [16,17]. A recent initiative to increase preparedness to climate risks is the Glasgow Climate Pact, where representatives from almost 200 countries met in November 2021 in the UN Climate Change Conference (CON26) and agreed on future strengthening of mitigation measures and new 2030 emissions targets (reducing methane emissions) and boosted efforts to deal with climate impacts [18,19]. In addition, other factors are the population aging, the ever-increasing rates of chronic diseases, high healthcare costs, misallocation of financial resources combined with inefficiency of health services and wastage of low-value healthcare, the pressures created by new medical technologies and drugs, lack of evidence to support reforms, etc. [13,20,21,22,23].

Health systems are essential in order to attain and preserve social health and well-being and are key factors to growth. It is becoming clear that appropriate organizational changes, health policy reformation, operational procedures, and management models should be made in order to achieve the sustainability of health systems and the health sector in general (sustainable healthcare). The WHO [24,25] defines Environmentally Sustainable Health as a system that “*improves, maintains or restores health, while minimizing negative impacts on the environment and leveraging opportunities to restore and improve it, to the benefit of the health and well-being of current and future generations*”.

According to Namany et al. [26], limited resources and unsustainable consumption could possibly result in ecological collapse and resource exhaustion, so it is imperative to have total resource efficiency in order to be sustainable so as to consume less resources and produce less waste but, at the same time, offer the same quality of services in the healthcare sector [27]. Researches have shown that healthcare providers have embraced and implemented various sustainability environmental practices, such as the adoption and use of green energy and the management and reduction of medical wastes [28,29,30,31]; specific pollution control (emissions to air, land, and water) [32,33,34,35,36]; practices for the protection, conservation and restoration of natural resources [5,37]; reuse/recycling, repair, and refurbishment of medical products [38,39]; efficient usage of resources [40]; sustainable procurement, etc. [1]. Therefore, healthcare providers, hospitals and healthcare systems, in general, need to alter their overall strategy in terms of sustainable development and through environmental sustainable transformation, so as to perform all the necessary actions, such as addressing and management of environmental risks, the adoption of the proper environmental and sustainable management policies and training, and the implementation of other initiatives in an efficient way in order to improve their reputation, reduce their operational costs and improve profitability because of the better energy efficiency, increase their staff’s satisfaction and retention, manage potential risks and comply with the legal framework in which they operate, and present a more eco-friendly and socially responsible image to its interested parties [41,42,43,44]. Moreover, it is essential to minimize and adequately manage HCWs and hazardous chemicals through the implementation of proper waste management methodologies, promote an efficient management of resources and sustainable procurements, introduce and monitor specific KPIs, reduce the health systems’ emissions of greenhouse gases and air pollution, engage the health workforce, and implement the respective tools in order to minimize the threats of the environment and to protect it [7,13,21,22,23]. Almost three years after the beginning of the pandemic of COVID-19 and its high rates of contagiousness, it is becoming more important for each healthcare system to have an effective and sustainable medical waste management system in order to protect public health [45].

## 2. Healthcare/Medical Wastes Quantities and Generation Rates

### 2.1. Indicative Evidence from Around the World

Healthcare is an ever-growing industry with a vast demand for healthcare services in various specialties that produce a large number of wastes, which can range from domestic to infectious or even radioactive, henceforth mentioned in this article as healthcare wastes (HCWs) [46,47,48,49].

According to the WHO [50], HCWs are all the waste that derive from the healthcare activities of hospitals, laboratories, research centers, mortuary and autopsy centers, animal research and testing laboratories, blood banks and collection services, and nursing homes for the elderly and can include a wide range of materials, such as used needles and syringes, soiled dressings, body parts, diagnostic samples, blood, chemicals, pharmaceuticals, medical devices, and radioactive materials. Almost 85% of this medical waste is general, non-hazardous waste, and the remaining 15% is considered hazardous waste and, more specifically, can be categorized as infectious, toxic, or radioactive [50,51,52].

It is becoming more clear that healthcare facilities around the world encounter many challenges and problematic situations such as the identification and evaluation of different categories of HCWs and the lack of a proper healthcare waste management system, regulations and its overall legislation, along with the necessary treatment technologies and methods which are used so as to dispose HCW; the implementation of specific practices from international public health organizations and agencies, such as the guideline for Safe management of wastes from health-care activities from WHO; and the provision of proper training for the staff responsible of the HCWs handling [12,52,53].

An international study from Singh et al. [30] regarding 24 countries with economies in transition shows that 18–64% of healthcare providers do not perform the appropriate HCW disposal techniques and do not possess a waste management system. In 2022, the WHO [54] highlighted that, at the international level, three out of ten healthcare facilities do not handle HCW safely, and this fact is more intense in regard to the least developed countries, where less than one out of three healthcare facilities have the basic services to handle HCW. In addition to that, studies claim that, before the pandemic of COVID-19, almost half the population of our planet was in danger due to environmental pollution and public health risks caused by the unsafe management and disposal of HCWs [30,55,56]. Moreover, the total number of HCW generated worldwide has a steady increase of 2–3% each year [51]. It is worth it to have a glimpse of what is happening around the world concerning the management of HCWs.

According to Khairunnisa et al. [3], India does not possess or implement a proper sustainable waste management system in its public healthcare sector. An average percentage of 10–15% of waste is hazardous and, in some cases, such waste is disposed of in the open environment (e.g., in the rivers); this could lead to the spread of toxic odors that could affect the public health due to the illnesses it can cause, such us hepatitis B and C, cholera, etc. [12,57]. Khalid et al. [58] state that the government teaching hospitals produced 900 kg/day medical wastes, government non-teaching hospitals produced 167 kg/day, and private teaching hospitals produced 79 kg/day. Altin et al. [59] state that the medical waste produced by four hospitals was 985 kg/day. Wassie et al. [60] state that African countries, especially Ethiopia, do not perform proper healthcare waste management practices, and the average rate of HCW generation in Africa is 0.8 kg/bed/day; Ethiopia has an average of 1.1 kg/bed/day alone [61,62]. Moreover, in Nigeria, the HCW generation is between 0.562 and 0.670 kg/bed/day, with a peak rate of 1.68 kg/bed/day [63,64,65]. Hamoda et al. [66] stated that the medical wastes produced by two public hospitals in Kuwait were 4.89–5.4 kg/day/patient and 3.65–3.97 kg/day/patient, respectively. In Tanzania, the HCWs produced from four hospitals are between 299 and 1554 kg/day [67]. A study of 57 hospitals in Lebanon [68] stated that private hospitals produce 2.45 kg/bed/day, and the other investigated hospitals produce 0.94 kg/bed/day.

Singh et al. [30] performed a survey to collect data from 78 countries, based on articles, publications, and other sources for the time period of January 2000 until May 2020, regarding the management of medical wastes worldwide. Their finding shows that most of the HCW is not managed in a correct way, and there is also an absence of proper training of the healthcare staff involved in the waste management process and safe practices [69,70]. Furthermore, they found that the average waste generation rate for the investigated countries is between 0.3 and 8.4 kg/bed/day, with the USA having the highest amount, at 8.4 kg/bed/day, and the health expenditure per capita is 9538.1$. Pakistan and Greece have the lowest amount of HCW, about 0.3 kg/bed/day, and the health expenditure per capita is 37.9$ and 1464.7$ respectively. Moreover, they found that approximately 67% is general waste, 27% is infectious or toxic waste, and approximately 4% sharps. Moreover approximately 40% of the healthcare workers were injured during the handling of medical wastes despite the fact that almost 41% of the responsible personnel had proper training.

Voudrias [71] stated that the US healthcare sector produces 5.9 million tons of waste each year, and Li et al. [72] stated that China will have approximately 2.496 million tons of HCW in 2023 [51]. It is crucial for the healthcare facilities to identify and report the type and quantity of HCWs they produce and manage because the disposal cost of hazardous wastes is almost ten times higher than that of the non-hazardous wastes [61,73].

This situation gets even worse if we take into account that the HCW produced during the last three years of the COVID-19 pandemic has increased dramatically, along with the lack of healthcare professionals that are needed to manage HCW but do not due to the increased demand for healthcare services during the pandemic; the pandemic is considered to have a vital impact to the environment and public health [63]. According to the United Nations Environment Programme [74], the pandemic has increased the amount of hazardous healthcare waste by 3.4 kg/bed/day, which is almost ten times more than the average volume of hazardous healthcare waste, which ranges from 0.2 to 0.5 kg/bed/day [50].

So, it is vital for the environment and also for the healthcare systems to reduce the HCW generation rate in the meta-covid period, and this will result in the reduction of the total amount of HCWs and also the related risks (sharp injuries, infection by pathogenic agents, chemical or radioactive contamination, diseases transmitted to the population, etc.). Suitable and effective regulations need to be mutually accepted and complied with on an international level.

Ranjbari et al. [51] performed extensive research regarding the various levels of HCW governance, especially in the European Union and identified the following levels:

Global regulation and Initiatives such as international agreements and conventions (e.g., Basel Convention on Hazardous Waste and Stockholm Convention on Persistent Organic Pollutants) and international organizations (e.g., WHO and International Atomic Energy Agency);

European Union Regulations, such as EU directives, strategies, and action plans (e.g., European Green Deal, Circular Economy Action Plan, EU Plastics Strategy, EU Chemicals Strategy for Sustainability, Waste Framework Directive, Directive on single-use plastics, Directive on waste electrical and electronic equipment (WEEE), European Directive 2008/98/EC, European Directive 2000/532/EC, European Directive 75/442/EEC, etc.);

Regional Cooperation Initiatives, such as EU Environmental Cooperation Programs (e.g., Economic and Investment Plan for the Western Balkans, Green Agenda in the Western Balkans, and Environmental Partnership Program for Accession);

National Regulation, such as regulations at the national level (e.g., laws, national policies, strategies, national action plans, guidelines, and national steering committees);

Local Communities, such as zero-waste NGOs, business sectors, local scientific communities, and other local stakeholders.

### 2.2. Evidence from Greece

Proper HCW management has been a priority for Greece in order to ensure and protect the environment and public health and to comply with the environmental policy of the European Union (EU) regarding the prevention and minimization of production and risk waste. Therefore, Greece set a National Waste Management Plan in 2020 which includes the management of hazardous healthcare wastes and aims to develop and implement a flexible, cost-effective, and effective needs management policy according to the peculiarities in the country [75].

According to the Joint Ministerial Decision 146163/2012 and the amendment of the 41848/1848/2017 (Governmental Gazette (FEK) 3649/Β/16-10-2017), the HCWs are defined as the waste generated by healthcare facilities and are mentioned in the waste list Annex to Commission Decision 2000/532/EC of 3 May 2000, as applicable [76]. HCWs are divided in two categories, the non-hazardous medical wastes (resembles household waste) and the hazardous medical waste, which are also divided in infectious waste, the mix hazardous waste, and the other hazardous waste [75]. To continue, incineration is considered to be the most appropriate way of treatment for HCWs because it decreases their weight and volume; however, pressure steam sterilization is more accepted by the community, in contrast with other waste-management-facility technologies [77,78,79,80].

Regarding the HCW generation in the Greek healthcare sector, the studies are limited [7,81,82,83,84,85,86,87,88]. Komilis et al. [52] and Voudrias [71] claimed that University Hospitals produce the largest quantities of medical waste (0.70 kg/bed/day), while private mental health clinics produce the smallest quantities (0.043 kg/bed/day), and, more significant, the military hospitals rank first in regard to toxic and infectious waste generation (0.68 kg/bed/day).

The Hellenic Environmental Inspectorate in Greece conducted a survey with questionnaires in 177 healthcare units, and according to the data, they produce an average 0.7 kg of hazardous waste per bed [89]. Zamparas et al. [90], in their study in which they developed a multicriteria model to examine available procedures, techniques, and methods of handling infectious waste of the Rio University Hospital, demonstrated that the average HCW generation in Greece is 1.4 kg/bed/day, while Singh et al. [30] stated that Greece has the lowest amount of HCW, about 0.3 kg/bed/day.

In 2018, 16,700 tons of hazardous HCW was produced from healthcare facilities, and from it, 12,800 tons was sterilized and 3900 tons was incinerated. Moreover, the Greek public healthcare facilities produced 9500 tons of hazardous HCW, which is, on average, 0.8 kg/bed/day. In addition, it is expected that the total HCW will reach 18,200 tons in 2025 and will exceed 18,800 tons in 2030 [75]. In the case of the Greek healthcare sector, in 2014, the healthcare gross emissions (MMtCO2e) were 4.1, and the healthcare emissions as % of national total were 3.7%; thus, immediate actions need to be taken by the Greek Health System to change the course toward zero emissions [91]. According to the Price Observatory of the Greek Ministry of Health [92], the waste-management cost of infectious HCW that is sterilized is 0.7 € per kg, and the waste-management cost of toxic infectious HCW that is incinerated is 1.7 € per kg.

Due to the limitations in the literature in Greece, this is the first study that includes HCWs’ data from all hospitals in the Greek public healthcare sector and their economic impact in the healthcare providers/public hospitals. Additionally, this study sought to investigate the factors that significantly affect the infectious waste in the Greek public hospitals, the association of hazardous waste quantity (in kilos) and the respective costs, and the provision of an average cost of hazardous waste management in the Greek public healthcare sector; most significant, this study attempted to provide specific prediction models which can predict the amount of HCWs that a healthcare facility produces, while taking into account certain factors.

## 3. Materials and Methods

### 3.1. Study Design, Sampling

Greece’s public healthcare sector is divided into seven health regions, with different types and numbers of hospitals in each of them, making a total of 121, which are presented in Table 1. In addition, the Greek public hospitals have been categorized into 5 types: small hospitals/health centers (with bed capacity up to 100), general hospitals, university hospitals, specialized hospitals of type I (which includes specialties such as ophthalmology, gynecology, pediatrics, etc.), and specialized hospitals of type II (cancer hospitals).

The main source of data was the online platform of the Greek Ministry of Health, “Business Intelligence System” (BI-Health), to which Greece’s public hospitals are obligated to submit operational and financial data each month. The “BI-Health” has a significant role in the organizational, operational, and financial modernization of the National Health System of Greece because it ensures the collection and process of detailed and aggregated data of the State’s Public Hospitals at a central operational level and allows for the dissemination of information to the management mechanisms with the utmost objective of improving the quality of health services provided. Some of them have a common administration and a common recording of their financial data on the BI platform, and this is why they were considered as one in our research. In some cases, we were required to cross-reference the data included in the BI, and for that reason, information from the official websites of the hospitals was used. Moreover, for the specific information considering the hazardous waste, an Excel file was sent to all Greek public hospitals on 2019 in which they filled in the total kilograms of infectious waste they produced for the year 2018, as well as the annual cost for their management.

In addition, individual-hospital characteristics that have been published in previous studies to affect infectious waste and many more that were available on the BI platform were screened to be included in our multivariable model as independent variables in order to assess their association with waste cost and quantity of hazardous waste in kilos [93,94,95,96]. The hospital’s type is the first characteristic that reflects major information about the size of each one and the number and the complexity of the cases it handles. The number of beds was also included in our suggested model, as it indicates its potential capacity. One more element that is expected to affect waste costs is the area in which each hospital is located. The separation recorded concerned the location of each hospital, in Mainland or Island Greece, thus indicating short or long distances from the special facilities for the management of infectious waste and, by turn, high or low transport costs [97,98]. To continue, the existence or not of special units, such as the Intensive Care Unit (ICU), Increased Care Unit, and Artificial Kidney Unit, was information that was included as well, because of the possible additional production of hazardous waste. Finally, the number of employees that serve as permanent staff and auxiliary staff was also examined.

The total annual healthcare activities performed in each hospital were also recorded and evaluated for a possible association with waste management costs and waste: The annual number of patients, internal and external, the number of laboratory tests (bio pathological tests, endoscopic examinations, and invasive diagnostic tests), medical Imaging tests, and others, both as a hole and individually, were recorded for the year 2018. Finally, the annual number of hemodialysis procedures performed in the Artificial Kidney Units and the total number of surgeries (both urgent and scheduled) were also examined for a possible association with hazardous waste quantities and their management costs.

### 3.2. Methods

For the analysis, linear models were used, with the management cost of waste and the kilos of waste serving as the dependent variables. Moreover, logarithms of the dependent variables, cost, and quantity, as they deviated from the normal distribution, were used, and applications and comparisons of several models were performed. First, ordinary least squares (OLS) were used [99,100]. In order to increase the efficiency of estimation, seemingly unrelated regression models were applied, a method proposed by Arnold Zellner [101]. Thus, we simultaneously computed separate regression models for each hospital’s waste cost, assuming that the (contemporaneous) errors associated with the dependent variables may be correlated. Having identified that the same set of independent variables is used for each dependent variable, a multivariate regression analysis was conducted in order to have a more complete picture. Since we had a large number of variables available for each hospital, the regression analysis was used to look for those variables that have statistically significant effects on the total cost of infectious waste management (*p*-value < 0.05). So, models for predicting both costs and quantities of waste with a coefficient of determination of R^2^ ≥ 0.85 were created, meaning that 85% or more of the variation in the cost or quantity of waste, respectively, is explained by the variables included in the models derived. Subsequently, in order to better appreciate how costs or quantities, respectively, are affected by the interacting variables in the models, adjusted estimates of the cost or quantity of waste were made, assuming that all other variables remain constant and at their 2018 values. Finally, by applying the 2018 data to the cost-projection models, the cost estimation for hospital waste management was calculated, which, in turn, was compared with actual hospital prices. All analyses were performed by using Stata version 13 [102], using the commands regress (reg), seemingly unrelated regress (sureg), and multivariate regression (mvreg). In all cases, significant results were considered to be those with a *p*-value < 0.05 (Figure 1).

## 4. Results

Our analysis revealed many differences in management costs and quantities of infectious waste generated by health region and/or by type of hospital, Figure 2 and Table 2. More specifically, in 2018, the 1st Health District, with a total of 22 healthcare hospitals, spent more than 1% of the total annual operating costs in the management of hazardous waste, and the 3rd Health District, consisting of 15 hospitals, spent the least amount, with 0.60%.

Significant differences are also observed in average costs per bed, per patient, among the Greek Health Districts and in average quantities of waste produced in different types of hospitals, as presented in Table 2, Table 3 and Table 4 and in Figure 3. Once again, in the 1st Health District, the cost of hazardous waste per bed is much higher than the average, thus suggesting, among other things, differences in the way infectious waste is managed in each health facility and in the Greek Health Districts in general, with or without infectious-waste management regulations, the level of trained staff, and other factors.

The variables that seem to have a statistically significant effect on waste management cost and hazardous waste quantities that are produced in Greek public hospitals are the number of beds, the hospitals type, the existence of an Intensive Care Unit, the number of the internal patients, their days of stay, and finally the number of the hospital’s employees. A strong positive correlation was found between all the variables with a *p*-value < 0.01. Table 5 indicates the most appropriate regression model to be used for our analysis, i.e., the seemingly unrelated regression model.

After repeated trials of the three regression analysis tests and excluding, each time, the independent variables that did not have statistically significant effects (*p*-values > 0.05) on the variables under study, the respective models are presented in Table 6.

The detailed results of the seemingly unrelated regression analysis that is the most appropriate for our data are shown in Table 7 and Table 8. The *p*-values < 0.05 were considered to be significant.

Based on the above results, the respective prediction models for the management costs and quantities of infectious waste for Greece’s public hospitals are presented (Table 9 and Table 10): A.LogWasteCost = 8.013 + 0.024 ∗ Beds + b2 ∗ HospitalType + b3 ∗ HospitalType ∗ Beds + b4 ∗ ICU + 0.0000265 ∗ Inpatients − 0.000006 ∗ Days + 0.0030473 ∗ Employees.B.LogWasteKilos = 6.711 + 0.0268 ∗ Beds + b2 ∗ HospitalType + b3 ∗ HospitalType ∗ Beds + b4 ∗ ICU + 0.0000337 ∗ Inpatients − 0.0000106 ∗ Days + 0.0024858 ∗ Employees − 0.0000331 ∗ S.Surgeries.

Both models have an R^2^ coefficient that is slightly greater than 85%, indicating that most of the variability in both management costs and the amount of hazardous waste generated is explained by their variables.

Considering the coefficients of the models for each of the independent variables, a positive correlation we observed between the number of beds and the total waste management costs, as well as the quantities of waste, with different weight depending on the type of each hospital. These two variables interact on our dependent variables under consideration in a way that is illustrated in Figure 4.

It is clear that the type of hospital imparts a different starting constant for the cost or quantity of waste (higher for general hospitals, university hospitals, and specialized type II hospitals (cancer hospitals) and lower for small hospitals and specialized type I hospitals (ophthalmology, gynecology, pediatrics, etc.) Figure 5.

However, it contributes inversely to these variables in conjunction with the number of beds of the hospitals. In small hospitals, waste management costs more as the number of beds increases, but in other hospitals, the rate of increase is much lower. In other words, the marginal cost, i.e., the extra cost of adding one more bed, is higher for small hospitals and lower for general, university, and cancer hospitals. The same trend appears to be followed by the quantities of waste generated in kilograms in relation to the additional beds for each type of hospital (Figure 6).

A positive contribution to the cost of waste management and to the quantities of waste generated can be observed from the number of hospitalized patients. On the contrary, the days of hospitalization have a very small but negative effect on the variables under study which can be logically explained as production of infectious waste for each patient decreases as the last days of hospitalization are approaching. The number of scheduled surgeries is seen to have a slightly negative effect on the quantities of waste generated, and this can be explained by the readiness of the staff in these surgeries as opposed to the state and/or level of organization during emergency surgeries. Finally, what seems to have a statistically significant positive effect on the variables under study is the number of staff working in hospitals, and this is quite logical since it is the employees, and specifically doctors and nurses, who manage the infectious waste of patients.

### Model Assessment

When calculating the predictions of waste management costs based on model A and the values of our independent variables for 2018, we found a better fit in the four categories of hospitals (general, university, and specialized types I and II) that actually manage much larger quantities of waste. In contrast, small hospitals were identified as having larger residuals in our predictions.

The larger deviations of the predicted values from the observed values in small hospitals that were found, as seen in Figure 7, can be related to the remote location where they are usually located, an element that is not taken into account by our models, as the only geographical categorization that was made concerned only continental or island regions. A large proportion of the country’s small hospitals are located on islands, but the rest of this category is located on the Greek mainland but in isolated areas, and this, in turn, leads to the more expensive transport of waste, costs that add to the overall contractual costs of hospital waste management. However, our models have a very good fit to the majority of public hospitals, and with the appropriate information additions, they can be improved even further.

## 5. Discussion

The study shows that the HCWs in the Greek public healthcare sector are affected by a number of individual-hospital characteristics such as the number of beds, the type of the hospital/healthcare provider, the services they provide, the number of annual inpatients, the days of staying, the number of scheduled surgeries, the existence of special units (e.g., Intensive Care Unit), and the total number of employees. Some of these factors are also mentioned and confirmed in various studies from different countries such as Taiwan, Jordan, Kuwait, India, and Nigeria, as well as from Greece, such as the type of the hospital, the number of inpatients–outpatients, etc. [83,86,98,103,104]. It is very possible that additional factors may significantly affect the production of HCWs, especially during the pandemic of COVID-19, where the HCWs’ generation rates and qualities were increased dramatically. It is an imperative need to extend the scope of this study into the Greek private healthcare sector in order to investigate the similarities and differences between them, as well as their correlations.

To continue, these factors that affect HCWs, along with the financial impact, can help hospital managements/managers, officials at the governmental level, and the Greek Ministry of Health to understand what needs to be altered and implemented in order to protect the environment, reduce hospital operational costs, and implement the necessary policies and action plan. For example, data show that the 1st Health District spends over 1% of its annual budget for the cost of hazardous wastes (which was almost 7,000,000 € in 2018), and with the proper actions, the amount could be reduced and reinvested in other areas of the healthcare sector.

According to our calculations with the available data from the Greek public hospitals in 2018, the total average waste cost per bed is 571.30 €, the total average waste cost per patient is 7.60 €, and the total average waste cost per day is 2.30 €. The number of beds in each hospital affects the quantities and costs of infectious waste management differently and according to the type of structure. In small hospitals, waste management costs more as the number of beds increases, but in other hospitals, the rate of increase is much lower. The hospitalized patients also have a positive correlation with both management costs and quantities of hazardous waste generated in each hospital, but the days of hospitalization have a very small but negative effect on them. Likewise, the number of scheduled surgeries have a slightly negative effect on the quantities of waste generated. Finally, the number of staff working in hospitals have a statistically significant positive effect on costs and quantities. It will be of great significance to compare the current results and findings with a future study, using up-to-date data from the Greek healthcare sector.

The rising of awareness and increasing of knowledge regarding HCW management, health, safety and environmental issues, and infectious waste risks should be performed by the hospital staff (especially doctors and nurses) via specialized training programs, posters, applied policies, lectures, etc. This will lead to the reduction of HCWs and the better segregation of regulated medical waste, consequently leading to the reduction of the volume and respective costs.

Furthermore, the two prediction models that were presented concerning the management costs and quantities of infectious waste for Greece’s public hospitals have an R^2^ coefficient slightly greater than 85%, thus indicating that most of the variability in both management costs and the amount of hazardous waste generated is explained by them. This enables the central administration of the Ministry of Health and of each hospital individually to identify deviations in costs and quantities of infectious waste, enabling preparation in terms of budgeting, evaluation, and improvement in the areas that are needed each time for better management in terms of sustainability, economics, and the environment.

In addition, these two prediction models can be used by governmental bodies and authorities in order to change the current status and improve the weaknesses, vulnerabilities, and the possible dysfunctions of the Greek public healthcare sector.

Thus, the following proposed actions have to be taken into consideration for implementation in the Greek healthcare sector so as to transform it into a sustainable one:A proper and customize sustainable waste management system should exist and function;Proper training regarding the waste management and the occupational safety of healthcare professionals;The implementation of a waste management policy and a customized Standard Operational Procedure (SOP);A review of the current legislation and relating policymaking;The implementation of new medical waste treatment technologies;Recycling of materials;The need for standardized and mutually accepted guidelines in national and international level of HCWs;A minimization of the costs and related risks from HCWs.

## 6. Conclusions

The present study ought to be the first completed and evidence-based attempt which was conducted for Greece and presents the status of HCWs in the public healthcare sector and its correlation with the financial impact. The models for predicting management costs and quantities of infectious waste that were presented can also be used as a model of health economic policies for cost management and as a tool to evaluate hospitals on infectious waste issues in order to optimize management and reduce costs in the healthcare sector.

It would be very interesting to see prospective studies conducted that investigate the HCW produced in the private healthcare sector in Greece and see if the factors affecting HCWs’ generation rates, quantities, and costs are similar. Furthermore, the methodologies and techniques that are used internationally, in terms of waste management, need to be investigated and presented in order to provide with an overall national strategy plan and present the best practices to reduce the total number of HCWs in the upcoming years, in respect to the protection and restoration of the environment, along with the sustainability of the Greek healthcare sector, with extended applicability to other healthcare systems.

Τhis practice can be applied in many countries that have similar data or even more of them. In this way, national or even international standards can be derived from meta-analyses of the factors influencing the quantities and costs of infectious waste management in hospitals. This means that there is the potential to derive appropriate limits on the quantities of infectious waste generated and, hence, on the costs of managing them, for each type of hospital separately and in relation to the number of beds and its operational dynamics. This, in turn, can form the backbone of national policies on infectious waste in public hospitals. It can serve as a tool to evaluate and improve practices in the management of infectious waste, which will be continuously improved with new data, thus making health systems more sustainable from an environmental and economic point of view.

## Figures and Tables

**Figure 1 ijerph-19-09821-f001:**
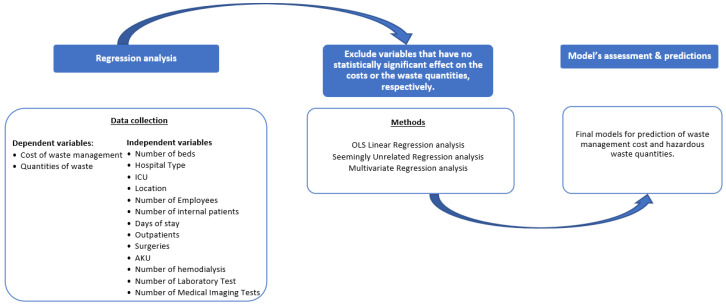
Schematic representation of the research methodology and the process of analysis.

**Figure 2 ijerph-19-09821-f002:**
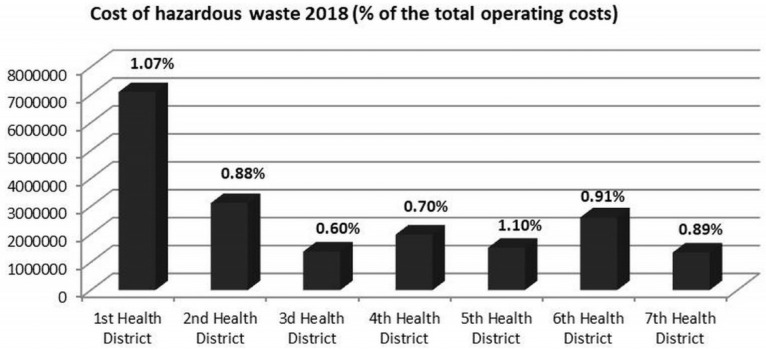
Cost (€) of hazardous waste management in 2018, per Health District in total and as a percentage of their total annual operating cost.

**Figure 3 ijerph-19-09821-f003:**
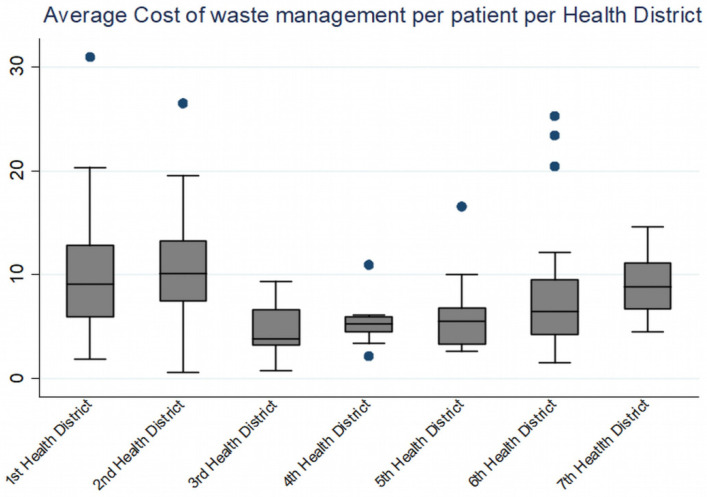
Average cost (€) of hazardous waste management for public hospitals in Greece in 2018 per patient and by health region.

**Figure 4 ijerph-19-09821-f004:**
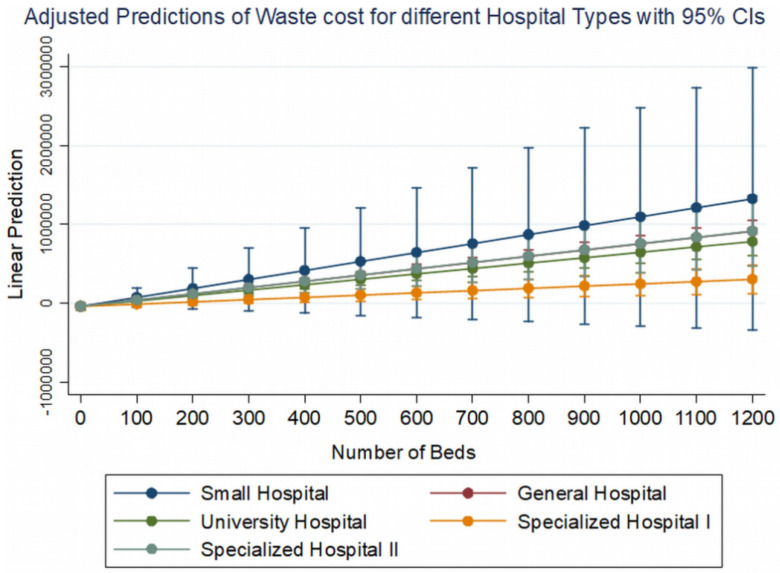
Adjusted predictions of waste management cost for different hospital types and for different number of beds based on model A, if all other variables remain constant. Please note that the adjusted predictions of general hospitals coincide with the adjusted predictions of specialized hospitals type II.

**Figure 5 ijerph-19-09821-f005:**
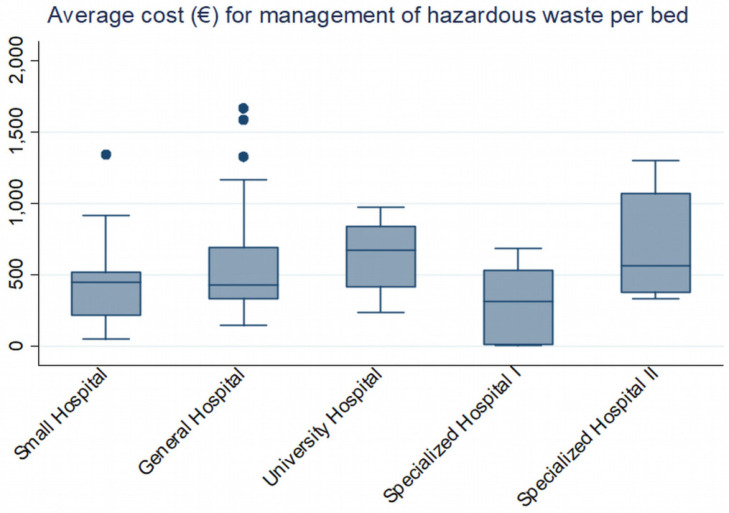
Average cost (€) of hazardous waste management for public hospitals in Greece per bed and per hospital type in 2018.

**Figure 6 ijerph-19-09821-f006:**
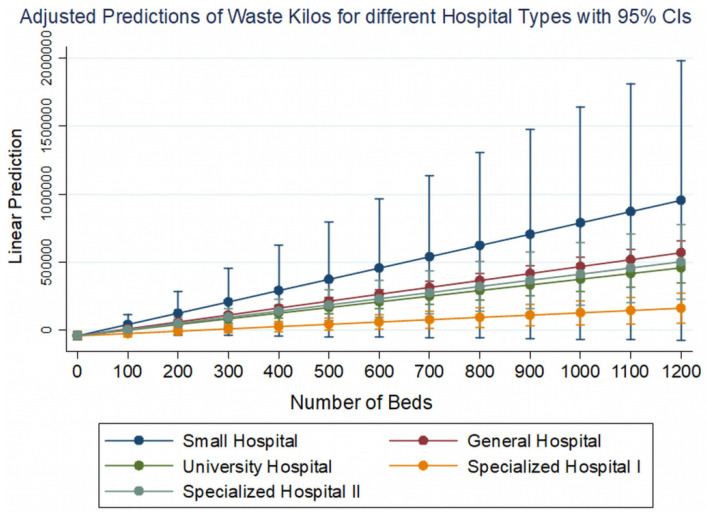
Adjusted predictions of waste quantities generated (kilos) for different hospital types and for different number of beds based on model B, if all other variables remain constant.

**Figure 7 ijerph-19-09821-f007:**
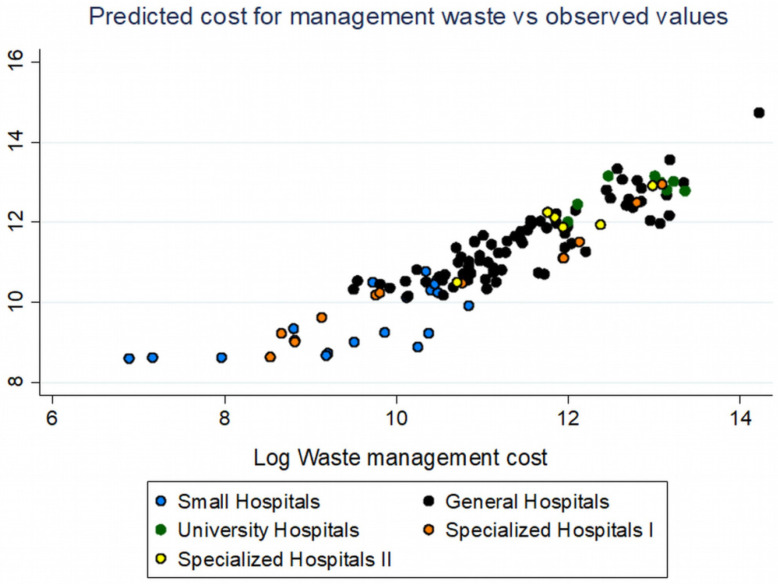
Predicted cost (logarithms) of waste management based on model A (*y*-axis) vs. observed values (logarithms) of cost for 2018.

**Table 1 ijerph-19-09821-t001:** Hospital distribution in Greece per health region and hospital type.

Health District	Small Hospitals	General Hospitals	University Hospitals	Specialized Type I Hospitals	Specialized Type II Hospitals
1st	1	11	0	7	3
2nd	6	9	2	2	2
3rd	0	14	0	1	0
4th	1	10	2	0	1
5th	2	10	1	0	0
6th	6	19	2	1	0
7th	3	4	1	0	0

**Table 2 ijerph-19-09821-t002:** Average annual costs of hazardous waste management in 2018 for Greece’s public hospitals (per bed) by Health District.

Health District	Average Annual Cost of Hazardous Waste per Bed (€)	Total Average Waste Cost per Bed (€)
1st	718	571.3
2nd	630	
3rd	355	
4th	395	
5th	467	
6th	449	
7th	601	

**Table 3 ijerph-19-09821-t003:** Average annual costs of hazardous waste management in 2018 for Greece’s public hospitals (per patient) by Health District.

Health District	Average Cost of Hazardous Waste per Patient (€)	Total Average Waste Cost per Patient (€)
1st	10.6	7.6
2nd	10.7	
3rd	4.4	
4th	5.3	
5th	6.1	
6th	8.2	
7th	9.0	

**Table 4 ijerph-19-09821-t004:** Average quantities in kilos of hazardous waste generated per bed and day of hospitalization in 2018 for public hospitals in Greece by hospital type.

Hospital Type	Average Kilos of Hazardous Waste per Bed per Day	Total Average Kilos of Hazardous Waste per Bed per Day
Small Hospital	0.33	0.84
General Hospital	0.75	
University Hospital	0.99	
Specialized Hospital I	0.34	
Specialized Hospital II	1.00	

**Table 5 ijerph-19-09821-t005:** Correlation table for the pairs of the statistically significant variables. High evidence of interdependencies (coefficient correlation > 0.5).

Pairwise Correlations	LogWaste Cost	LogWaste Kilos	Beds	Hospital Type	ICU	Inpatients	Days of Stay	Employees
LogWaste Cost	1.0000							
LogWaste Kilos	0.9492	1.0000						
Beds	0.6084	0.6790	1					
Hospital Type	0.2557	0.3017	0.3784	1.0000				
ICU	0.7271	0.7326	0.5579	0.1627	1.0000			
Inpatients	0.7544	0.7712	0.8005	0.3044	0.6509	1.0000		
Days of stay	0.6586	0.6961	0.9078	0.3149	0.5753	0.899	1.0000	
Employees	0.7759	0.799	0.924	0.3154	0.6436	0.8798	0.9217	1.0000

**Table 6 ijerph-19-09821-t006:** Regression analysis results. Coefficient of determination, R^2^, and statistically significant variables from each regression method for estimating management costs and the amount of infectious waste generated in Greek public hospitals.

Dependent Variables	YEAR 2018					
	Seemingly Unrelated Regression		Multivariate Regression		Linear Regression	
	Number of Obs = 121		Number of Obs = 121		Number of Obs = 121	
	R^2^	Independent Variables	R^2^	Independent Variables	R^2^	Independent Variables
Cost of hazardous waste	0.8522	Beds, Hosptype, Beds#Hosptype Icu, Inpatients, Days, Employees	0.8603	Beds, Hosptype, Beds#Hosptype Icu, Inpatients, Days, Employees	0.8522	Beds, Hosptype, Beds#Hosptype Icu, Inpatients, Days, Employees
Kilos of hazardous waste	0.8594	Beds, Hosptype, Beds#Hosptype Icu, Inpatients, Days, Employees, S.Surgeries	0.8642	Beds, Hosptype, Beds#Hosptype Icu, Inpatients, Days, Employees	0.8471	Beds, Hosptype, Beds#Hosptype Inpatients, Days, Employees

**Table 7 ijerph-19-09821-t007:** Statistically significant variables (coefficients and their standard errors) with seemingly unrelated regression for prediction of cost for hazardous waste management in Greek public hospitals.

Seemingly Unrelated Regression for the Cost of Hazardous Waste Management		R^2^ = 0.8522	
		Number of Obs = 121	
		Coefficient	(SE)
Constant	b0	8.013451	(0.2663484)
Number of Beds	b1	0.0246254	(0.0053255)
Hospital Type			
General Hospital	b2	2.118334	(0.2903447)
University Hospital		5.640858	(1.147168)
Specialized Hospital Type I		1.613669	(0.3829162)
Specialized Hospital Type II		1.867596	(0.6772875)
Hospital Type # Beds			
General Hospital	b3	−0.0287761	(0.0052831)
University Hospital		−0.0352248	(0.0055343)
Specialized Hospital Type I		−0.0278102	(0.005303)
Specialized Hospital Type II		−0.0276302	(0.0057483)
Intensive Care Unit	b4	0.4388912	(0.1553799)
Total Internal Patients	b5	0.0000265	(6.85 × 10^−6^)
Days of Stay	b6	−0.00000633	(2.83 × 10^−6^)
Total Number of Employees	b7	0.0030473	(0.0003703)

**Table 8 ijerph-19-09821-t008:** Statistically significant variables (coefficients and their standard errors) with seemingly unrelated regression for prediction of quantities of hazardous waste generated in Greek public hospitals.

Seemingly Unrelated Regression for the Kilos of Hazardous Waste		R^2^ = 0.8594	
		Number of Obs = 121	
		Coefficient	(SE)
Constant	b0	6.711418	(0.293857)
Number of Beds	b1	0.0268199	(0.0058768)
Hospital Type			
General Hospital	b2	2.482767	(0.3203334)
University Hospital		5.751815	(1.26892)
Specialized Hospital Type I		1.803676	(0.4244299)
Specialized Hospital Type II		2.805956	(0.7482615)
Hospital Type # Beds			
General Hospital	b3	−0.027856	(0.0058301)
University Hospital		−0.0341217	(0.0061111)
Specialized Hospital Type I		−0.027567	(0.0058515)
Specialized Hospital Type II		−0.0285752	(0.006342)
Intensive Care Unit	b4	0.3689101	(0.1717678)
Total Internal Patients	b5	0.0000337	(7.68 × 10^−6^)
Days of Stay	b6	−0.0000106	(3.19 × 10^−6^)
Total Number of Employees	b7	0.0024858	(0.0004086)
Scheduled Surgeries	b8	−0.0000331	(0.293857)

**Table 9 ijerph-19-09821-t009:** Values of model A’s coefficients b2 and b3 for each hospital type.

Hospital Type	b2	b3
Small Hospital/Health Center	0	0
General Hospital	2.118334	−0.0287761
University Hospital	5.640858	−0.0352248
Specialized Hospital Type I	1.613669	−0.0278102
Specialized Hospital Type II	1.867596	−0.0276302

b4 = 1 when there is an ICU and 0 when there is not one.

**Table 10 ijerph-19-09821-t010:** Values of model B’s coefficients b2 and b3 for each hospital type.

Hospital Type	b2	b3
Small Hospital/ Health Center	0	0
General Hospital	2.482767	−0.027856
University Hospital	5.751815	−0.0341217
Specialized Hospital I	1.803676	−0.027567
Specialized Hospital II	2.805956	−0.0285752

b4 = 1 when there is an ICU and 0 when there is not one.

## Data Availability

No new data were created or analyzed in this study. Data sharing is not applicable to this article.

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
