# Peer review of "Identifying and Predicting Healthcare Waste Management Costs for an Optimal Sustainable Management System: Evidence from the Greek Public Sector"

_ijerph, 2022, doi:10.3390/ijerph19169821_

Round 1
Reviewer 1 Report
Line 2.2 This section “Methods” is one of the most important ones of an article. However, in your article, this section is extremely short. “Coefficient of Determination” is not mentioned nor is “adjusted predictions” or “predicted cost”. It will be very helpful to include a scheme of the methodology used with all the stages that you performed for this study.
Line 323. Figure 1. Please write the name of the axis X, Y, and the units for Y.
Line 361. Table 6. Use lower cases for the information presented in this table.
The discussion section is weak. Comparing your results with scientific literature related to this subject is necessary.
Line 482 says “from our experience and knowledge” what are you talking about? Please be more specific.
For the conclusions section, it would be desirable to have a wider vision, not only a conclusion for Greece, but I would also suggest concluding how to use this article in other countries.
Author Response
Reviewer 1
1) Line 2.2 This section “Methods” is one of the most important ones of an article.
However, in your article, this section is extremely short. “Coefficient of Determination” is not mentioned nor is “adjusted predictions” or “predicted cost”. It will be very helpful to include a scheme of the methodology used with all the stages that you performed for this study.
[FR]: Some additional text and clarifications about the method used are added. Please advice if there is anything else that needs to be done.
Please refer to p. 7, section 3.2. Methods, the text in red color.
2) Line 323. Figure 1. Please write the name of the axis X, Y, and the units for Y.
[FR]: It is corrected.
Please refer to p. 8, Figure 2.
3) Line 361. Table 6. Use lower cases for the information presented in this table
[FR]: It is corrected.
Please refer to p. 11, Table 6
4)The discussion section is weak. Comparing your results with scientific literature related to this subject is necessary.
[FR]: Additional text, clarifications and references were added in this sector. Please advise if there is anything else that needs to be done.
Please refer to p. 16, the text in red color.
5)Line 482 says “from our experience and knowledge” what are you talking about? Please be more specific.
FR]: It is rephrased.
6) For the conclusions section, it would be desirable to have a wider vision, not only a conclusion for Greece, but I would also suggest concluding how to use this article in other countries.
[FR]: Additional text was added in this sector. Please advise if there is anything else that needs to be done.
Please refer to p. 17, the text in red color.
Reviewer 2 Report
The study presents the status of healthcare wastes in the public Greece healthcare sector and its correlation with the financial impact. The models for predicting management costs and quantities of infectious waste that was arrived can also be used as tools to evaluate hospitals on infectious waste issues and thus to optimize management and reduce costs. Large discrepancies between the estimates and actual values of the hospitals under consideration can highlight mismanagement or even good practices. It would be very interesting in conducting prospective studies that will investigate the HCW produced in the private healthcare sector in Greece and the methodologies-techniques they use in terms of waste management in order to provide with an overall national strategy plan and present the best practices to reduce the total number of HCWs in the upcoming years, given respect to the protection and restoration of the environment along with the sustainability of the Greek healthcare sector.
This is valuable paper and should be published after minor revision as below.
General Remarks
- Abstract
Please do not use abbreviation in abstract or provide the full name.
- Introduction is too long, especially first part. It should be decreased by 20-25%
- Number of References is very high (98) and some of these are not strictly connected with the subject area. It should be decreased by30-40%.
- References should not to be used in Conclusions
Detailed remarks
1. Lines 256-280. Some references should be added
2. 2.2. Methods. Some references on statistic software should be added
3. Figure 1, 2, 3, 4, 5, 6 should be self-explaining. Please add proper information.
4. Table 5 -10 should be self-explaining. Please add proper information.
5. Line 504. I propose to remove …”We believe that”…
Author Response
Reviewer 2
General Remarks
1) Abstract|: Please do not use abbreviation in abstract or provide the full name.
[FR]: It is corrected.
2) Introduction is too long, especially first part. It should be decreased by 20-25%
[FR]: We have broken down the introduction section in two parts: 1. Introduction (page 1) & 2. Healthcare/Medical Wastes quantities & generation rates (page 3) [with two sub-sections: 2.1 Indicative evidence from around the world (page 3) & 2.2 Evidence from Greece (page 5)]. Please advise if this is acceptable.
3) Number of References is very high (98) and some of these are not strictly connected with the subject area. It should be decreased by30-40%.
[FR]: On behalf of all researchers of this paper, I have to point out that all references mentioned are specific and on topic. We believe that it is not a good idea to reduce our selected references and if do so, it will affect the final deliverable. In any case, if there are some references that you think that should not be included in this paper, please let us know in detail so as to delete them. In addition, more references have been added. Please refer to pages 23-24, the text in red color.
4) References should not to be used in Conclusions.
Detailed remarks
[FR]: It is corrected.
Please refer to p. 17, section 6. Conclusions, the text in red color.
- Lines 256-280. Some references should be added .
[FR]: We do not understand where exactly we need to add some further reference. Please advise.
- 2.2. Methods. Some references on statistic software should be added.
[FR]: It is corrected.
Please refer to p. 7, section 3.2. Methods, the text in red color.
- Figure 1, 2, 3, 4, 5, 6 should be self-explaining. Please add proper information.
[FR]: It has been added. Please check the names of Figures in red color
- Table 5 -10 should be self-explaining. Please add proper information.
[FR]: It has been added. Please check the names of Figures in red color
- Line 504. I propose to remove …”We believe that”… [
FR]: It is rephrased.
Reviewer 3 Report
The manuscript identifies the main challenges and current situation of the Healthcare Waste Management sector with Greece as a case study.
The topic is quite current especially due to the COVID-19 pandemic. I have some comments that would be beneficial to the paper before publication:
1. Introduction section is far too long. It would be best to divide this section in two parts for instance, introduction and background or something similar.
2. It would be good to mention at the end of the introduction the sections that will follow (structure of the paper).
3. It is necessary to explain how the results obtained and the methods used can be applied elsewhere? What is their significance on a worldwide basis? How can these results be translated into policy?
4. Regarding styling, there is no need to add the default title on top of the graphs, the legend below is enough.
5. Waste is better used as a singular noun except where it defines a specific area of waste i.e. hospital wastes.
6. It would be best to avoid using the 'we' person in the manuscript but instead use the third person overall.
7. If an acronym is defined, no need to re-use the whole text see line 47, 108.
8. It would be best to have the paper proofread as there are some English errors throughout.
Author Response
Reviewer 3
1. Introduction section is far too long. It would be best to divide this section in two parts for instance, introduction and background or something similar.
[FR]: We have broken down the introduction section in two parts: 1. Introduction (page 1) & 2. Healthcare/Medical Wastes quantities & generation rates (page 3) [with two sub-sections: 2.1 Indicative evidence from around the world (page 3) & 2.2 Evidence from Greece (page 5)]. Please advise if this is acceptable.
2. It would be good to mention at the end of the introduction the sections that will follow (structure of the paper).
[FR]: The proposal was good but given respect to the format that the sections were written, we believe that it is not necessary to present the structure of the paper as text.
3. It is necessary to explain how the results obtained and the methods used can be applied elsewhere?
[FR]: It is corrected.
Please refer to p. 7, section 3.2. Methods, the text in red color.
What is their significance on a worldwide basis? How can these results be translated into policy?
FR]: Additional text was added in the sector “Conclusions” which we hope covers the questions.
Please refer to p. 17, section 6. Conclusions, the text in red color.
4. Regarding styling, there is no need to add the default title on top of the graphs, the legend below is enough.
[FR]: The titles are not defaulted by Stata, the statistical package we used to create them. We added them intentionally to make the graphs more comprehensible. For this reason we would like them to remain as they are.
5. Waste is better used as a singular noun except where it defines a specific area of waste i.e. hospital wastes.
[FR]: They have been corrected.
6. It would be best to avoid using the 'we' person in the manuscript but instead use the third person overall.
[FR]: They have been corrected.
7. If an acronym is defined, no need to re-use the whole text see line 47, 108.
[FR]: They have been corrected.
8. It would be best to have the paper proofread as there are some English errors throughout
[FR]: They have been corrected.
Round 2
Reviewer 1 Report
Well done Authors. Your MS has improved significantly.